# VH3-53/66-Class RBD-Specific Human Monoclonal Antibody iB20 Displays Cross-Neutralizing Activity against Emerging SARS-CoV-2 Lineages

**DOI:** 10.3390/jpm12060895

**Published:** 2022-05-29

**Authors:** Sergey V. Kulemzin, Maria V. Sergeeva, Konstantin O. Baranov, Andrey A. Gorchakov, Sergey V. Guselnikov, Tatyana N. Belovezhets, Olga Yu. Volkova, Alexander M. Najakshin, Nikolai A. Chikaev, Daria M. Danilenko, Alexander V. Taranin

**Affiliations:** 1Institute of Molecular and Cellular Biology, Siberian Branch of Russian Academy of Sciences, 630090 Novosibirsk, Russia; skulemzin@mcb.nsc.ru (S.V.K.); baranov@mcb.nsc.ru (K.O.B.); gorchakov@mcb.nsc.ru (A.A.G.); sguselnikov@mcb.nsc.ru (S.V.G.); belovezhec@mcb.nsc.ru (T.N.B.); volkova@mcb.nsc.ru (O.Y.V.); najakshin@mcb.nsc.ru (A.M.N.); na_chik@mcb.nsc.ru (N.A.C.); 2Smorodintsev Research Institute of Influenza, 197376 St. Petersburg, Russia; maria.sergeeva@influenza.spb.ru (M.V.S.); daria.danilenko@influenza.spb.ru (D.M.D.)

**Keywords:** SARS-CoV-2, COVID-19, variants of concern, Omicron, neutralizing monoclonal antibodies, mutational escape, VH3-53/66, VH1-58, iB14, iB20

## Abstract

Immune evasion of SARS-CoV-2 undermines current strategies tocounteract the pandemic, with the efficacy of therapeutic virus-neutralizing monoclonal antibodies (nAbs) being affected the most. In this work, we asked whether two previously identified human cross-neutralizing nAbs, iB14 (class VH1-58) and iB20 (class VH3-53/66), are capable of neutralizing the recently emerged Omicron (BA.1) variant. Both nAbs were found to bind the Omicron RBD with a nanomolar affinity, yet they displayed contrasting functional features. When tested against Omicron, the neutralizing activity of iB14 was reduced 50-fold, whereas iB20 displayed a surprising increase in activity. Thus, iB20 is a unique representative of the VH3-53/66-class of nAbs in terms of breadth of neutralization, which establishes it as a candidate for COVID-19 therapy and prophylactics.

## 1. Introduction

Antibodies preventing cellular entry of SARS-CoV-2 are considered pivotal in protective immunity against coronavirus infection. Most SARS-CoV-2-neutralizing antibodies are known to act by blocking the interaction of the viral spike (S) receptor-binding domain (RBD) with its receptor, human ACE2. The lack of effective virus-targeting therapies has spurred massive efforts to identify and extensively characterize potent neutralizing monoclonal antibodies as a measure to counteract the COVID-19 pandemic. To date, about 3000 SARS-CoV-2-targeting nAbs have been obtained by multiple research groups across the globe. The vast majority of such nAbs were selected to recognize the ancestral Wuhan-1 spike or an early, very similar, viral variant B.1 [1]. These efforts have culminated in the approval of several nAbs or nAb cocktails as prophylactics or early intervention therapeutics in high-risk groups of patients.

In mid-2020, novel SARS-CoV-2 viral variants started to emerge. Some of these variants, such as B.1.351/Beta, P.1/Gamma or B.1.617.1/Kappa, had little success in leaving the regions where they werefirst reported, whereas others, such as B.1.1.7/Alpha and B.1.617.2/Delta, have spread globally. As it turned out, these emerging SARS-CoV-2 variants displayed increased resistance to immunity elicited by the ancestral Wuhan-1 or B.1 viral variants. For instance, Beta and Gamma variants share three key amino acid substitutions in their RBD (K417N/T, E484K и N501Y) and are poorly neutralized by the antisera from some of the previously infected or vaccinated donors [2,3]. Furthermore, these variants are partially or completely resistant to neutralization by most of the nAbs identified early in the pandemic by screening against wild-type or nearly wild-type spike/RBD antigens [4,5].

In late 2021, Omicron sublineage BA.1 was first described in South Africa. This variant features an unprecedented number of mutations and high transmissibility, which ensured its rapid spread around the globe. BA.1 Sprotein differs from the ancestral sequence by 6 deletions, 3 insertions and 30 single amino acid substitutions, of which 15 are found in the RBD. Clearly, then, this virus is resistant to a significant proportion of antibodies elicited by the vaccination or infection with other viral lineages [6,7,8].

The ability of SARS-CoV-2 to evade humoral immunity is a serious challenge for programs aiming to introduce vaccine- and nAb-based countermeasures against the pandemic. Notably, at the time of writing, due to the massive spread of nAb-resistant SARS-CoV-2 variants, the FDA has revoked or revised its emergency use authorizations for seven out of the eightnAbs approved earlier. Therefore, the identification of broadly neutralizing Abs (bnAbs) capable of withstanding the emerging viral lineages of SARS-CoV-2 continues to be a pressing issue of vital importance.

Previously, our group reported the identification of a panel of ultra-potent SARS-CoV-2-specific human nAbs [9]. Two of the antibodies from this panel, iB14 and iB20, have been demonstrated to neutralize a number of circulating viral variants including Alpha, Beta, Delta, Kappa, and Lambda. Based on the structure of their VH-gene sequences, iB14 and iB20 belong to two different clonotypes referred to as VH1-58- and VH3-53/66-classes. The fact that VH1-58-class nAbs can neutralize various SARS-CoV-2 variants was also demonstrated by other research groups [10,11]. However, the cross-neutralizing activity of iB20 was quite surprising to us, given the observation that most of the VH3-53/66-class of nAbs displayed rather limited neutralization breadth due to their sensitivity to the widely spread RBD substitutions K417N, N501Y, or E484K [5,6,7,12,13]. Here, we studied the binding and neutralizing properties of nAbs iB14 and iB20 towards Omicron (BA.1). We have shown that, whereas neutralizing the potency of iB14 drops ~50-fold, iB20 neutralizes Omicron even better than it does B.1.

## 2. Materials and Methods

### 2.1. Cloning, Antibody Expression and Purification

The pCAGGS-omiSpikeΔ19 construct encoding the spike of the Omicron variant of SARS-CoV-2 lacking 19 C-terminal amino acid residues was obtained by replacing the sequences of the ancestral Wuhuan-1 spike in the pCAGGS-SpikeΔ19 plasmid [9] with a gene-synthesized (Genewiz, Chelmsford, MA, USA) CDS of the Omicron spike, using the XbaI and NotI cloning sites. Similarly, the sequences encoding heavy and light chain variable fragments of the nAb CV30 [14] were extracted from PDB (6XE1) and synthesized by Genewiz. A VH-encoding sequence was then inserted between the AgeI and SalI restriction sites of the pAbVec-HC construct in frame with the leader sequence and the human IgG1 constant region sequence, whereas a VK sequence was placed between the AgeI and BsiWI sites of the pAbVec-KC plasmid.

For antibody expression, HEK293T cells were transfected with pairs of expression constructs encoding appropriate light and heavy chains using the calcium phosphate transfection protocol. Typically, 200 μg of each plasmid was used to transfect 10^8^ HEK293T cells growing in a 5-layer cell factory (Corning™ Falcon™). Eight hours following transfection, the growth medium was replaced with a serum-free medium, EX-CELL^®^ 293 Serum-Free Medium (Sigma, St. Louis, MO, USA). Six days later, cell supernatants were collected and filtered through a 0.22 μm PES-filter (TPP), and antibodies were purified on a Protein A agarose column (McLab, San Francisco, CA, USA, #PPA-503). Briefly, after the loading step, the column was washed twice with PBS. Then, the antibodies were eluted with 0.1 M glycine, pH 2.7, 150 mMNaCl, into the neutralization buffer of 1 M Tris, pH 8.0, in a 1:10 ratio. Antibody preparations were then dialyzed against three changes of PBS and concentrated using Ultra-15 Ultracel-100K (Amicon, Miami, FL, USA) to a final concentration of 1–2 mg/mL.

### 2.2. SARS-CoV-2 Pseudovirus Neutralization Assay

The production of S-pseudotypedlentiviral particles and pseudovirus neutralization assays has been described in our earlier study [9]. Briefly, HEK293T cells were transfected with a 4:6:3 molar mixture of plasmids psPAX2, pCDH-NLuc, and a pCAGGS-SpikeΔ19 plasmid encoding either a WT or a BA.1 variant of the truncated SARS-CoV-2 S protein (SΔ19). After 8 h of incubation, the growth medium was replaced with Opti-Mem (Gibco, Thermo Fisher Scientific, Waltham, MA, USA) and supplemented with 2.5% heat-inactivated FBS (Gibco, Thermo Fisher Scientific). Supernatants were collected 48 h later, 0.45 μm filtered and concentrated by centrifugation at 20,000× *g*, 4 °C for 90 min. Lentiviral particles obtained were used for the transduction of ACE2-HEK293T cells in the presence of different concentrations of neutralizing antibodies. Antibodies were serially diluted in Opti-MEM+2.5% FBS in twofold steps to concentrations ranging from 0.5 ng/mL to 1 µg/mL and co-incubated with 20,000 S-pseudotypedlentiviral particles for 30 min at 37 °C prior to transduction. Non-transduced ACE2-HEK293T cells and those transduced in the absence of antibodies served as the controls. Forty-eight hours following transduction, the luminescence was measured in the cultures using aLuminoskanluminometer (Thermo Fisher Scientific) and anNLuc FLASH Assay (NanoLight^®^ Technologies, Pinetop, AZ, USA). The half-maximal inhibitory concentration (IC50) was determined by non-linear regression as the concentration of antibody that neutralized 50% of the pseudotypedlentivirus. The data from two independent experiments were used.

### 2.3. Authentic SARS-CoV-2 Virus Neutralization Assay

We followed the cell ELISA-based virus-neutralization assay adapted for coronaviruses SARS-CoV-2 [15] and OC43 [16], this format was chosen due to its simplicity and high reproducibility in our hands.

Two SARS-CoV-2 isolates were used: hCoV-19/Russia/StPetersburg-3524/2020 (B.1 sublineage) and hCoV-19/Russia/StPetersburg-6086/2022 (BA.1.1 sublineage).

Vero cells (ATCC #CCL-81) were seeded in 96-well plates at 2.5 × 10^6^ cells/well 24 h before the neutralization assay. On the day of the experiment, two-fold antibody dilutions starting with 10 μg/mL were prepared in cell culture medium. Next, virus (100 TCID50/50 μL) aliquots in a total volume of 150 μL were added to the equal volume of antibody dilutions. The mixture was incubated for 1 h at 37 °C in U-bottom plates (Medpolimer). Culture medium was removed from the cells, followed by a single wash with a serum-free medium. Antibody/virus mixtures were then added to the cells (100 μL/well in duplicate). Forty-eighthours later, the medium was aspirated from the wells, and the cells were fixed in 80% acetone for 10 min at room temperature. Following fixation, the cells were washed twice with 500 μL PBS-0.1% TWEEN20 (PBS-T) using an automated microplate strip washer ELX50 (BioTek) and blocked overnight in 5% non-fat dry milk in PBS-T, 200 μL/well at +2–+8 °C. The next day, the wells were washed twice with PBS-T, and the cells were incubated with SARS-CoV-2 N-specific mouse monoclonal antibodies (Xema-Medica, Kiev, Ukraine) for 1.5 h at room temperature. The cells were then washed four times with PBS-T and stained with HRP-conjugated goat anti-mouse IgG (Abcam, #ab97040), 1:5000 in blocking solution, 100 μL/well for 1 h at room temperature. Following 6 washes with PBS-T, TMB substrate (Bioservis) was aliquoted to the wells (100 μL/well). The reaction proceeded for 7 min at room temperature in the dark and was terminated by the addition of 100 μL/well 1N H_2_SO_4_ (Vekton). Optical density (OD) was measured on a ClarioStarmicroplate reader (BMG Labtech), and the resulting OD value was calculated by subtracting the background OD (655 nm) from the specific OD (450 nm) values.

### 2.4. BiolayerInterferometry

Binding kinetics of the recombinant iB20 and iB14 nAbs with RBDs from SARS-CoV-2 Wuhan-1, Omicron (BA.1) and SARS-CoV were analyzed on an Octet RED96 instrument (ForteBio, Pall Life Sciences, New York, NY, USA). Samples were run in 1×ForteBio buffer (PBS, 0.05% Tween, 0.05% NaN_3_), stirring speed 1000 rpm, total volume 200 μL/well at 30 °C in flat-bottom 96-well plates (Greiner Bio-One, Littleton, CO, USA). Antibodies were immobilized on anti-human Fc capture biosensors for 300 s. Association events with RBD dilutions (36 nM, 18 nM and 9 nM) were monitored for 600 s, followed by a dissociation step for 900 s. To correct for baseline drift, average shift values observed for the sensor incubated with RBD in the absence of antibody immobilization were subtracted. BLI data were processed using ForteBio Data Acquisition Software v.11.1.1.19 (1:1 model with global fitting). Pairwise competition assays for the binding of iB20 and CV30 were performed on NTA biosensors. After Wuhan RBD was immobilized on the sensor via its 6 × His tag (200 s), the first antibody (600 s) was added, followed by the second (600 s).

## 3. Results

### 3.1. iB14 and iB20 Display Reduced Binding to Omicron RBD

The resistance of the Omicron variant to convalescent sera and monoclonal nAbs is largely attributable to multiple amino acid substitutions in its S protein, with the RBD alone having 15 substitutions (G339D, S371L, S373P, S375F, K417N, N440K, G446S, S477N, T478K, E484A, Q493R, G496S, Q498R, N501Y, Y505H). To understand whether these changes may affect the binding of iB14 and iB20, we produced recombinant RBD of Omicron and Wuhan-1. As assayed by BLI, iB20 associates with Omicron RBD with a KD = 4.5 × 10^−9^, which is ~10-fold lower compared to the interaction with the canonical RBD sequence (KD = 0.57 × 10^−9^) yet is still within the range reported for other potent nAbs (Figure 1). This reduction in affinity is mostly due to the higher dissociation rate of the RBD/nAb complex (Koff 3.72 × 10^−4^ vs. 3.35 × 10^−5^, respectively), with the association rate being slightly higher (Kon 8.2 × 10^4^ vs. 5.9 × 10^4^). The affinity of the interaction between iB14 and Omicron RBD was also reduced compared with that of the wild-type RBD (KD = 6.7 × 10^−9^и 0.73 × 10^−9^, respectively). In this case, however, this was caused by the slower complex formation (Kon 6.0 × 10^4^ (Omicron RBD) vs 1.8 × 10^5^ (w.t. RBD)). As a control, we also measured the interaction between iB14/iB20 with the RBD of SARS-CoV, a close relative of SARS-CoV-2. These two viruses share several conserved S epitopes, which can be recognized by cross-neutralizing antibodies [17]. Our data indicate that iB14 and iB20 target the epitopes unique to SARS-CoV-2, as no interaction with the SARS-CoV RBD was observed.

### 3.2. iB20 nAbIs A Potent Neutralizer of Omicron

Based on the encouraging BLI data, iB20 and iB14 were expected to also retain their neutralization activity against Omicron. To test whether this was the case, we first turned to a pseudovirus neutralization assay and used lentiviral particles pseudotyped with SARS-CoV-2 Wuhan-1 or Omicron (BA.1) S protein. iB14 was found to display a ~50-fold reduction in neutralization potency, which dropped from an IC50 3 ± 1 ng/mL to 174 ± 47 ng/mL (Figure 2A). This is in line with the observations made for the VH1-58 class of nAbs that iB14 belongs to [6,8]. In striking contrast, iB20 displayed an order of magnitude stronger neutralization activity (110 ± 43 ng/mL (Wuhan-1) vs. 18.4 ± 7.5 ng/mL (Omicron)) (Figure 2A).

Next, we proceeded to neutralization assays using the authentic SARS-CoV-2 viruses, namely Omicron and B.1 lineage representatives (B.1 Spike differs from that of the ancestral Wuhan-1 by a single amino acid substitution, D614G, found outside the RBD and having no influence whatsoever on the binding of RBD-specific mAbs). In this assay, iB14 was a weak neutralizer of the Omicron virus (IC50 = 10 µg/mL), whereas, again, iB20 neutralized Omicron better than it did B.1. This ~2-fold improvement in neutralization was consistent with our pseudovirus neutralization assay data (Figure 2B).

### 3.3. iB20 Competes for RBD Binding with VH1-53/66-Class Member, nAb CV30

The observation that iB20 has cross-neutralizing activity was puzzling, as this feature is not characteristic of VH3-53-class nAbs. One explanation was that, despite the use of the VH3-53 gene and sensitivity to S substitutions N460T and A475V, iB20 binds the region of RBD that is distinct from the epitope targeted by other VH3-53/66-class nAbs. To test this possibility, we performed competition binding of iB20 and CV30 to RBD, as the latter nAb is a typical representative of the VH3-53/66-class. Recombinant CV30 was produced in-house based on the published sequences [14]. BLI experiments were performed with an RBD pre-immobilized on the biosensor, followed by the consecutive addition of iB20 or CV30 nAbs. Regardless of the order of nAb presentation to the biosensor, binding kinetics were very similar (Figure 3). Namely, the addition of the second nAb did not result in a further signal increase, which indicates that iB20 and CV30 binding epitopes significantly overlap.

## 4. Discussion

Of thousands of SARS-CoV-2-specific nAbs identified to date, only a small proportionare known to neutralize the Omicron variant of SARS-CoV-2 [6,7,8,18]. Typically, they do so at a higher concentration, compared to the neutralization of earlier viral variants. For instance, sotrovimab, which was, until recently, one of the most efficient cross-neutralizing antibodies approved for therapeutic applications [17,19], requires a 3–7 times higher concentration to neutralize Omicron than it does when tested against the B.1 variant [6,7,8,20]. Two notable examples of this rule currently include bebtelovimab and romlusevimab, which received approval for use in high-risk groups of patients [20,21,22].

The VH1-58 class representatives that displayed potent cross-neutralization against all pre-Omicron SARS-CoV-2 variants [9,10,11] have largely failed against Omicron [6,8,18,20]. iB14, identified and characterized earlier by our group, was no exception, and displayed a 20- to 50-fold reduced neutralization potency against authentic Omicron or Omicron S-pseudotypedlentiviral particles compared to the B.1 variant. Structural studies may be needed to address the mechanism causingthis tooccur, as our experiments indicate that, despite the 10-fold decrease in the affinity of iB14 to Omicron RBD, it is still within a nanomolar range, which is typically sufficient for very potent neutralization. We speculate that, in the context of authentic Omicron virus, iB14 interaction with its cognate epitopes may be sterically hindered.

VH3-53/66-class nAbs are among the most frequent anti-SARS-CoV-2 antibodies, constituting about 20% of the antibodies reported in the CoV-AbDab [1,23]. Depending on the length of CDRH3, they form two distinct clonotypes [12,13,23]. Typically, nAbs having a short CDRH3 ranging 9–13 amino acid residues share a described or a predicted mechanism of S binding mediated by the germline residues of CDRH1 and CDRH2 [12,13]. The footprint of such nAbs within RBD extensively overlaps with the region of RBD interaction with ACE2. As a rule, VH3-53/66-class nAbs do not display cross-neutralizing properties, because their epitopes frequently include S amino acid residues K417 and/or N501 that are mutant across multiple SARS-CoV-2 lineages such as Alpha, Beta, Gamma, and Omicron. These substitutions result in a highly attenuated binding of VH3-53/66-class nAbs and adversely affect their neutralization potency. Only a fewnAbs from this class, including those isolated from a patient infected with a Beta-variant of SARS-CoV-2, have been reported to be resistant to K417N/T and/or N501Y effects [6,24]. Nonetheless, to our knowledge, the vast majority of nAbs from this class display no or very weak neutralization of the Omicron variant [6,7,25].

On the one hand, iB20 is a typical representative of the VH3-53/66-class of nAbs. Competitive binding BLI experiments indicate that VH3-53 nAb CV30 [24] and iB20 epitopes overlap. Similarly to other members of this class, iB20 is sensitive to S substitutions N460T and A475V [9,13,26]. CDRH1 and CDRH2regions of iB20 encompass the motifs 33NY34 and 53SGGS56, which have been previously established to be critical for RBD recognition by the VH3-53/66-class of nAbs [12]. On the other hand, iB20 displays a breadth of neutralization that is not characteristic of the VH3-53/66-class of nAbs. In our previous work, iB20 was demonstrated to neutralize a broad range of SARS-CoV-2 variants, which we attribute to its ability to withstand the deleterious effects of K417N/T, N501Y, and E484K [9]. Here, we observed a puzzling 2-fold improvement in neutralization that iB20 shows against Omicron, compared to the nearly ancestral B.1 variant. Structural features underlying such activity are currently unclear. The affinity of the iB20 towards Omicron variant RBD is one order of magnitude lower than it is for the Wuhan-1/B.1 RBD. This higher potency is likely attributable to the higher on-rate of nAb/RBD complex formation. However, much as with iB14, the major contributing factor in authentic virus neutralization is related to the steric details of nAb interaction with S trimers on the surface of viral particles. Given that iB20 is virtually identical to other VH3-53 nAbs in the framework regions, its unique functional features are most likely defined by the CDRH3 and VL sequences. In this regard, resolving the structure of iB20 complexed with various S variants is highly warranted, as this information would be instrumental for structure-driven modification of other nAbs from this class, as well as for designing vaccine antigens that are tailored for bnAb induction. Finally, the cross-neutralizing properties of iB20 establish it as a potential candidate for COVID-19 prophylactics and therapy.

## Figures and Tables

**Figure 1 jpm-12-00895-f001:**
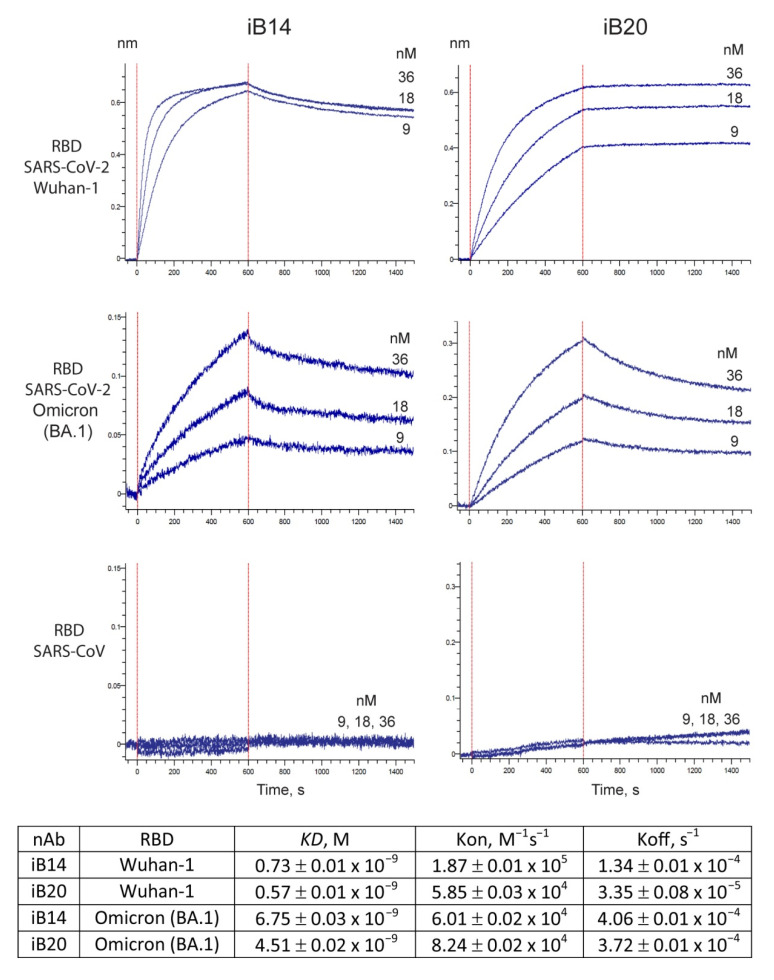
BLI analysis of the interaction between nAbs iB14 and iB20 (9–36 nM) with the indicated RBDs of SARS-CoV-2 and SARS-CoV.

**Figure 2 jpm-12-00895-f002:**
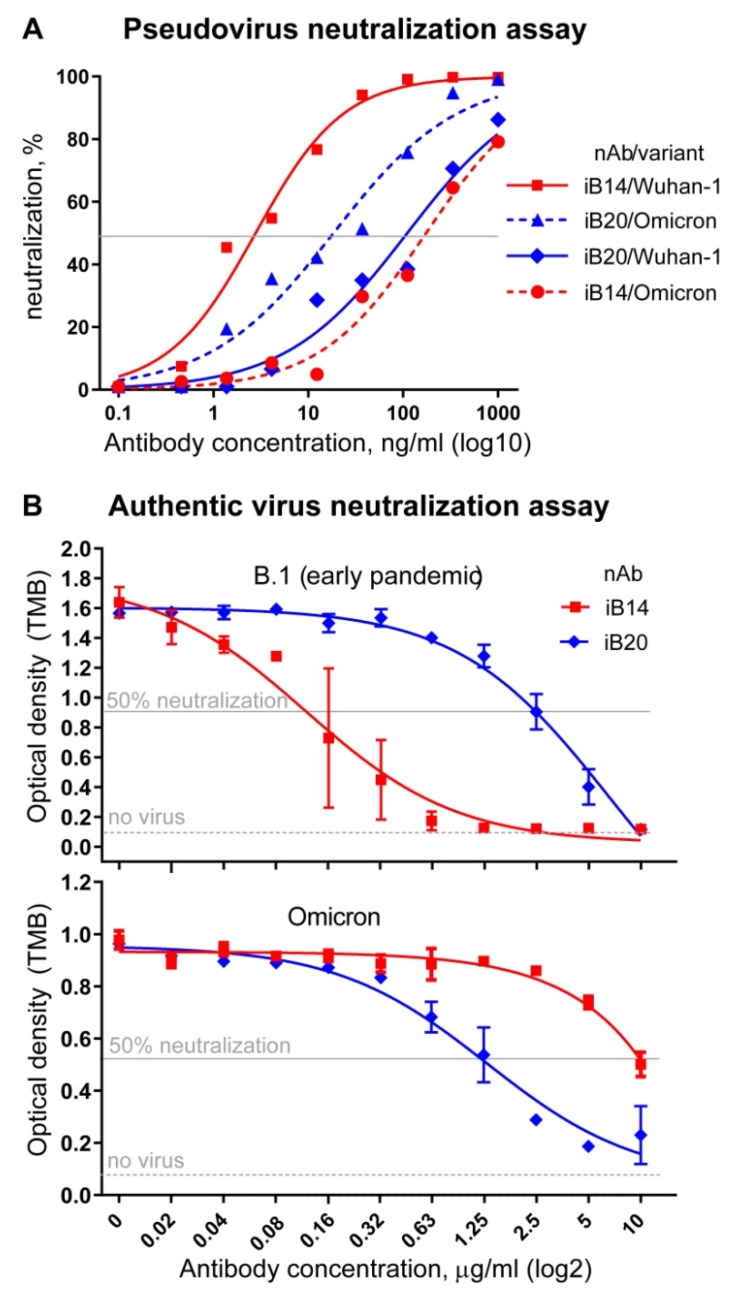
Activity of iB14 and iB20 in pseudovirus (**A**) and authentic SARS-CoV-2 (**B**) virus-neutralization assays. Early pandemic (Wuhan-1 or B.1) and Omicron variants were used.

**Figure 3 jpm-12-00895-f003:**
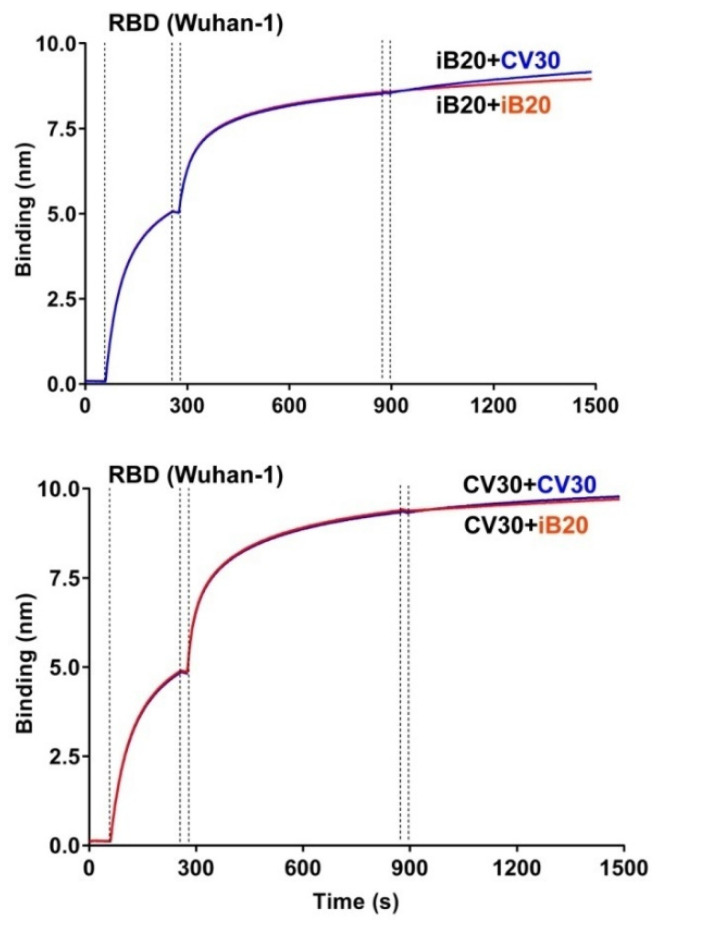
Competition BLI assay for iB20 and CV30 binding to SARS-CoV-2 RBD.

## Data Availability

Not applicable.

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
