# Peer review of "VH3-53/66-Class RBD-Specific Human Monoclonal Antibody iB20 Displays Cross-Neutralizing Activity against Emerging SARS-CoV-2 Lineages"

_jpm, 2022, doi:10.3390/jpm12060895_

Round 1

Reviewer 1 Report

In the manuscript “VH3-53/66-class RBD-specific human monoclonal antibody iB20 displays cross-neutralizing activity against emerging SARS-CoV-2 lineages”, Kulemzin and collaborators describe the interesting results they obtained when two monoclonal antibodies to the SARS-CoV-2 spike they had previously described in earlier work were tested with the Omicron variant. As expected, one had decreased efficacy. Surprisingly, however, the other monoclonal antibody was better at binding to and neutralizing the omicron variant than an earlier variant. I believe that the manuscript addresses an important scientific and technological question, given that antibodies are a crucial therapeutic tool in Covid-19. Here are some comments I have regarding the manuscript.

Major issues:

1 – One of the key methods used in the manuscript is the “authentic SARS-CoV-2 virus neutralization assay”. Instead of using plaque formation as readout, the authors did a colorimetric assay with an antibody that detects the SARS-CoV-2 N antigen. It makes comparing these results with those of other manuscripts in the literature harder, and many readers might not be familiar with it. It is important to show or cite other papers that show that this methodology accurately reflects neutralization and how its results compare with titers from plaque-forming assays.

2 – On lines 180-181, the authors state that “Recombinant CV30 was produced in-house based on the published sequences”. The materials and methods section should explain how this reagent was produced and purified.

Minor issues

3 –Instead of describing the methodology of the pseudotyped lentivirus assays, they just reference the previous manuscript. I would suggest the authors include at least a brief description of the methods and plasmids used in the present manuscript.

4 – On the introduction section, the authors make multiple references to websites. I would suggest they might not be necessary.

5 – Lines 40 – 43: “As it turned out, these emerging SARS-CoV-2 variants displayed pronounced resistance to immunity elicited by the ancestral Wuhan-1 or B.1 viral variants.” I believe this is overstating, as clinical studies showed that previous infection and especially vaccinations with booster shots, all made with Wuhan-1 sequences, were clearly still protective against all variants so far. There was definitely some loss in protection, but not “pronounced resistance”.

6 – Similarly, on lines 43 – 44: “Beta and Gamma variants […] are poorly neutralized by the antisera from previously infected or vaccinated donors”. Again, this might be an overstatement. Studies definitely showed a decrease in neutralizing titers, but these titers were still significant in convalescent plasma.

7 – Lines 52 – 53: “Clearly then, this virus is uniquely resistant to antibodies elicited by vaccination or infection with other viral lineages”. Again, I would argue “uniquely resistant” might be an overstatement, as the authors themselves point that “several [nAbs] are known to neutralize the Omicron variant” (lines 192-193) and the vaccines were still effective.

8 – The manuscript is generally well written, but a few little issues remain that could be revised. For example: Line 86: 2.5*106 cells (6 is not superscript). Line 88: ug instead of µg. Line 113: “Geiger Bio-One” instead of Greiner Bio-One.

9 - Line 192 says “Of hundreds of SARS-CoV-2 specific nAbs identified to date”, whereas lines 29-30 says “To date, about 3000 SARS-CoV-2 targeting nAbs were selected”.

Reviewer 2 Report

The study is conducted on identifying cross-reactive mAbs that neutralize Omicron variant. This work is the need of the day. After some minor spell corrections, I endorse the findings in this study.
